# Open-Door ICU Model and Humanized Care: A Systematic Review

**DOI:** 10.3390/nursrep15110406

**Published:** 2025-11-18

**Authors:** Paula Andrea Duque, Sara Quintero Duque

**Affiliations:** School of Nursing, University Catholic of Manizales, Manizales 170001, Colombia; sara.quintero@ucm.edu.co

**Keywords:** family, hospice care, intensive care unit, nursing care, patient

## Abstract

**Background:** Management of patients in closed-door intensive care units (ICUs) is often associated with limited family visits and a highly technological environment, which can lead to patient deconditioning through altered circadian rhythms and depersonalization, contributing to psychological distress in addition to physiological distress. In recent years, there has been a shift in trends in the management of ICU patients with an emphasis on more social and psychological support, with the option of an open-door ICU. **Objective:** This study aims to evaluate the role of humanized care through social and psychological support in improving patients’ outcomes through the concept of open-door ICUs. **Methods:** This systematic review was conducted under the Preferred Reporting Items for Systematic Reviews and Meta-Analyses (PRISMA 2020) guidelines. Six databases were searched (LILACS, SciELO, PubMed, Scopus, ScienceDirect, and Dialnet) using a strategy based on MeSH and DeCS terms. Studies published between 2018 and 2025, in English, Spanish, and Portuguese, were included. Fifty studies were selected and analyzed using open, axial, and selective coding techniques. The review protocol was registered in PROSPERO (CRD420251080952). **Results:** Three main categories emerged: (1) Patient and Family Perceptions of ICU Care; (2) flexible visitation and technological mediation; and (3) humanization-centered care. These factors were linked to improved emotional well-being, reduced anxiety, enhanced communication, and stronger trust between families and healthcare professionals. **Conclusions:** Based on the results of our systematic review, we emphasize the importance of adopting humanized care practices in open-door ICUs. In particular, strategies like flexible visitation, emotional and spiritual support, respectful communication, and family involvement contribute to compassionate, patient-centered care. We recommend institutional policies that need to be designed that support humanization for patients and families.

## 1. Introduction

The concept of the open-door intensive care unit (ICU) reflects a growing global movement toward family-centered and humanized critical care. This model challenges traditional restrictive visitation policies by promoting open access for family members, encouraging their active participation in patient care, and fostering transparent communication with healthcare teams. Open ICUs aim to reduce patient stress, support emotional well-being, and strengthen trust between families and professionals. As this approach gains global relevance, it is essential to examine its impact on care quality, staff experience, and patient recovery [1].

Open ICUs also facilitate nonverbal communication strategies that enhance interaction with intubated or sedated patients. This approach promotes more person-centered care by increasing family participation and strengthening both communication and psychosocial support, in contrast to conventional intensive care practices that tend to prioritize technological intervention and physiological monitoring [1,2,3].

Although modern ICUs have evolved to provide more personalized and continuous care, environmental factors such as intense lighting, alarm noise, and frequent staff conversations may still make these environments feel unwelcoming to some patients and families, despite ongoing efforts to reduce sensory overload and thereby prevent delirium. Such environmental conditions can contribute to confusion, anxiety, and emotional vulnerability, as well as increase the risk of depersonalization and dehumanization within closed systems [3,4].

Restricted visitation and the inability of family members to accompany patients often lead to stress and a perceived loss of control. Traditional ICU models reflect a persistent biomedical approach that tends to instrumentalize the patient, prioritizing disease management over the recognition of individuals with thoughts, emotions, and relational needs [5,6]. Implementing an open-door ICU model represents an evidence-informed strategy to counter these limitations by fostering family participation, emotional support, and transparency in care processes, in line with contemporary frameworks for humanized intensive care [7,8]. Open-door policies in ICUs have been associated with reduced stress, anxiety, fear, and depression, as well as improved coping with the emotional challenges of hospitalization [9,10,11,12,13,14,15].

Humanized care in the ICU involves delivering kind and empathetic treatment during procedures, recognizing and alleviating pain, addressing family needs, and respecting the dignity and privacy of each patient. It seeks to achieve therapeutic goals without causing unnecessary [3,4,5,6]. Implementing an open-door ICU model constitutes a practice that enhances transparency, emotional support, and family participation in care, aligning with current standards for humanized intensive care. However, despite international efforts, the adoption of this model remains inconsistent, and its effects on healthcare professionals, patients, and families are not yet fully understood.

Objective: to evaluate the role of humanized care through social and psychological support in improving patients’ outcomes through the concept of open-door ICU.

## 2. Materials and Methods

### 2.1. Study Design, PICO Framework, and Eligibility Criteria

This systematic review was conducted following the Preferred Reporting Items for Systematic Reviews and Meta-Analyses (PRISMA 2020) guidelines [16]. The process comprised the following stages: (1) identifying the research question, (2) conducting a comprehensive literature search, (3) selecting studies according to predefined inclusion and exclusion criteria, (4) appraising methodological quality, and (5) synthesizing and reporting the results.

P—Population: Healthcare professionals, patients, and their families in Intensive Care Units (ICUs).

I—Intervention: Implementation of the open-door ICU model.

C—Comparison: Traditional/closed-door ICU models or absence of open-door policies.

O—Outcome: Factors facilitating or hindering the delivery of humanized care.

The research question was: “What are the key factors that influence the delivery of humanized care during the implementation of the open-door ICU model, as experienced by healthcare professionals, patients, and their families?”

Inclusion criteria were:(1)Studies addressing the implementation or experience of the open-door ICU model.(2)Research on humanized care in intensive care settings.(3)Studies describing strategies to enhance patient and family participation in ICU care.(4)Inclusion of healthcare professionals, patients, or families.(5)Publications from 2018 to 2025.(6)English, Spanish, or Portuguese language.(7)Pediatric and/or adult ICU settings.

Exclusion criteria were:(1)Studies limited to architectural or logistical aspects without a humanized care component.(2)Editorials, opinion papers, letters, and conference abstracts without full text.(3)Studies with unclear or inadequate methodology.(4)Full texts unavailable due to access restrictions.(5)Duplicate publications.(6)Books or non-peer-reviewed sources.

### 2.2. Information Sources and Search Strategy

A comprehensive literature search was conducted in LILACS, SciELO, PubMed, Scopus, ScienceDirect, and Dialnet to identify relevant studies published between 2018 and 2025. The search strategy combined controlled vocabulary from MeSH and DeCS with free text terms, including Intensive Care Units, ICU, Critical Care, Open-door ICU, Open-door model, Humanized Care, Family involvement in ICU, Health Professionals, Patients, and Family.

Boolean operators (AND, OR) and the wildcard symbol (*) were used to combine terms and retrieve variations across databases. The following representative search equation was applied:

((“Intensive Care Units” [MeSH] OR “ICU” OR “Critical Care”) AND (“Open-door policy” OR “Open door ICU” OR “Flexible visitation” OR “Family-centered care”) AND (“Therapeutic restrictions” OR “Conflict” OR “Humanized care” OR “Patient-centered care”)).

No restrictions were placed on study design, allowing the inclusion of qualitative, quantitative, and mixed methods studies, provided they addressed the implementation, experiences, or outcomes related to the open-door ICU model and its contribution to humanized care.

To ensure transparency and reproducibility, the complete search strategies for each database, including the exact dates of the last search, are provided in Appendix A. Searches were carried out between 3 April and 24 June 2025, and no limits were placed on study design or publication year.

### 2.3. Selection Process

All records identified through database searches were imported into Rayyan QCRI (Qatar Computing Research Institute) for management and duplicate removal. The selection process followed the PRISMA 2020 recommendations and was carried out in two stages: (1) title and abstract screening, and (2) full-text review.

Two independent reviewers screened all records against the predefined inclusion and exclusion criteria. The reviewers worked in double-blind mode within Rayyan to ensure transparency and minimize selection bias. Discrepancies were discussed and resolved through consensus with a third reviewer.

Full-text studies that met the eligibility criteria were included in the final synthesis. Reasons for exclusion at the full-text stage were documented, and the overall selection process is summarized in the PRISMA 2020 flow diagram (Figure 1).

### 2.4. Data Collection Process

Data extraction was conducted independently by three reviewers using a standardized form developed in Microsoft Excel. For each included study, the following information was extracted: author(s), title, year of publication, country, study design, research approach, methods, population, main findings, and thematic concepts.

The reviewers worked independently and in duplicate, and discrepancies in extracted information were discussed until consensus was reached. When necessary, data were verified through cross-checking with the full text of the articles to ensure accuracy and completeness.

### 2.5. Data Synthesis

A thematic synthesis approach was applied to integrate and interpret the findings. Qualitative coding was conducted in three stages: open, axial, and selective coding to identify recurrent themes and organize them into conceptual categories.

Descriptive matrices were developed to support comparison across studies, allowing for the identification of convergent and divergent perspectives on the open-door ICU model and its relationship with humanized care. The results were synthesized narratively and grouped into three main analytical categories: (1) Patient and Family Perceptions of ICU Care; (2) flexible visitation and technological mediation; and (3) humanization-centered care.

### 2.6. Data Items

For each included study, data was extracted for both descriptive variables and outcome domains relevant to the review question.

Descriptive variables included: Author(s), year of publication, and country. Study design and methodological approach. Setting (ICU characteristics) and participant groups (patients, relatives, or healthcare professionals). Type of intervention or policy (open-door ICU, flexible visitation, humanization programs). Funding source or institutional context (if reported).

Outcome domains focused on variables directly related to the review objectives, such as: Perceptions and experiences of humanized care among patients, families, and professionals. Effects of open-door or flexible visitation policies on emotional well-being, communication, and satisfaction. Facilitators and barriers influencing the implementation of open-door ICU practices. Ethical, cultural, and organizational implications of humanization in critical care.

All reported results compatible with these domains were extracted, regardless of the outcome measures or time points used in each study. When data were unclear or partially reported, information was inferred from contextual descriptions or cross-checked against other sections of the article (methods and discussion). No contact was made with study authors to obtain missing data, as most studies provided sufficient contextual detail for synthesis.

### 2.7. Study Risk of Bias Assessment

The methodological quality and risk of bias of the included studies were appraised using the Joanna Briggs Institute (JBI) Critical Appraisal Tools [17], applying the appropriate checklist for each study design (qualitative, quantitative, mixed methods, or review-based). Two reviewers independently conducted the appraisal, and discrepancies were resolved through consensus. Each criterion was rated as “met,” “not met,” or “unclear,” and the overall quality level (high, moderate, or low) was determined based on the proportion of criteria fulfilled.

Among the 50 included studies, methodological quality was predominantly high. Review-based studies (38%) demonstrated moderate-to-high methodological quality (mean compliance 81%), with some variability attributed to limited transparency in methods and absence of bias assessment in narrative and integrative reviews. Cross-sectional studies (30%) showed strong methodological soundness (mean compliance 87%), with clear inclusion criteria and appropriate analytical approaches, indicating a low risk of bias. Qualitative studies (18%) exhibited consistently high quality (mean compliance 88%), characterized by methodological congruence and rich description of participants’ perspectives, although reflexivity was occasionally underreported. Randomized controlled trials (8%) and the quasi-experimental design (2%) demonstrated robust rigor (mean compliance 86%), ensuring reliable intervention assessment. Mixed-methods (2%) and cohort/longitudinal studies (2%) achieved full or near-full adherence to JBI criteria, reflecting strong methodological consistency.

Overall, 54% of the studies included were rated as high quality, 36% as moderate, and 10% as low, indicating a generally low risk of bias and satisfactory methodological rigor across the evidence base. These results strengthen the reliability of the synthesized findings. A detailed summary of the JBI appraisal by study design is presented in Table 1.

### 2.8. Effect Measures

Given the qualitative and descriptive nature of the studies included, no statistical effect measures were calculated. Quantitative findings were summarized using descriptive statistics (frequencies, percentages, and mean values when available), while qualitative results were synthesized narratively through thematic analysis.

### 2.9. Synthesis Methods

Quantitative data were summarized in tables and analyzed descriptively to highlight patterns related to visitation policies, patient family communication, and institutional practices. Qualitative data were analyzed using the constant comparative method, following grounded theory principles [18]. Three reviewers analyzed the data independently and discussed findings until consensus was reached, ensuring analytical rigor and reflexivity.

Following this process, results from individual studies were compared, grouped, and integrated to identify convergent and divergent perspectives. A thematic synthesis approach was used to develop overarching conceptual categories that reflected the relationship between the open-door ICU model and humanized care.

Through iterative comparison, three final analytical themes emerged: (1) Patient and Family Perceptions of ICU Care; (2) flexible visitation and technological mediation; and (3) humanization-centered care.

### 2.10. Reporting Bias Assessment

No formal assessment of publication or reporting bias was conducted, as the included studies were predominantly qualitative or descriptive and did not provide comparable quantitative data suitable for meta-analysis. However, potential bias was minimized through comprehensive search procedures, including multiple databases in three languages (English, Spanish, and Portuguese).

All inclusion and exclusion decisions were transparently documented, and no evidence of selective reporting or outcome omission was identified during data extraction.

### 2.11. Certainty Assessment

No formal assessment of certainty was performed because the included studies were heterogeneous in design and primarily qualitative, descriptive, or review-based. Instead, confidence in the synthesized results was established through an integrative evaluation of methodological rigor, consistency, and coherence across studies.

The credibility of the evidence was supported by the high methodological quality scores obtained using the Joanna Briggs Institute (JBI) Critical Appraisal Tools and by the thematic convergence observed among the included studies. Divergent findings were examined through continuous comparison and interpretative triangulation to ensure a balanced synthesis.

Overall, the level of confidence in the body of evidence was considered high, as most studies demonstrated strong methodological soundness, low risk of bias, and consistent thematic patterns linking the open-door ICU model with humanized care practices and family-centered approaches.

### 2.12. Registration and Protocol

This systematic review followed a predefined protocol that was registered in PROSPERO (International Prospective Register of Systematic Reviews) under the identification number CRD420251080952. The review process adhered to the registered protocol, and no major deviations were made regarding the research question, eligibility criteria, or synthesis approach.

## 3. Results

### 3.1. Study Selection

A total of 943 records were retrieved from six databases. Before screening, 37 duplicate records, 415 ineligible records, and 216 records removed for other reasons were excluded. After these exclusions, 275 records (SciELO = 60, ScienceDirect = 27, PubMed = 82, Scopus = 31, LILACS = 50, and Dialnet = 25) were screened based on title and abstract. Of these, 120 records were excluded, leaving 155 reports assessed for eligibility. Following full-text review, 46 studies were excluded for lack of full text, 29 were editorials or opinion papers, 24 were conference proceedings, and 56 were not related to the theme. Finally, 50 studies met all inclusion criteria and were included in the final synthesis. The overall selection process is illustrated in the revised PRISMA 2020 flow diagram (see Figure 1).

### 3.2. Study Characteristics

The 50 studies included were conducted between 2018 and 2025 in more than 20 countries. Most were from Colombia (18%), Spain (16%), Chile (8%), Ecuador (8%), and other regions, such as the United States, the United Kingdom, and Mexico (See Table 2).

Review studies (integrative, narrative, systematic, and scoping): 19 studies (38%), representing the largest group and reflecting the growing interest in synthesizing existing evidence on humanization strategies in intensive care units (ICUs). Cross-sectional studies (descriptive, correlational, or comparative): 15 studies (30%), focused on exploring perceptions, experiences, and associated factors among healthcare professionals, patients, and families. Qualitative studies (phenomenological, descriptive, or interpretive): 9 studies (18%), providing an in-depth understanding of lived experiences and meanings related to humanized care in critical settings. Randomized controlled trials (RCTs): 4 studies (8%), offering evidence on the effectiveness of interventions aimed at improving communication and family participation in ICU care. Cohort or longitudinal studies: 1 study (2%), analyzing changes or effects over time. Quasi-experimental design: 1 study (2%), evaluating pre- and post-implementation effects of specific interventions.

Mixed-methods design: 1 study (2%), integrating both qualitative and quantitative approaches within a single research framework to provide a comprehensive understanding of humanized care (see Table 2).

### 3.3. Risk of Bias Within Studies

The methodological quality of the included studies was assessed using the Joanna Briggs Institute (JBI) Critical Appraisal Checklists, tailored to each study design. Overall, most studies demonstrated high methodological quality, with a mean score of 86% across all designs. qualitative studies and procedural detail in quantitative designs. Quality ranged from 80% to 100%, with minor reporting gaps related to researcher reflexivity in qualitative studies and procedural detail in quantitative designs.

A complete summary of individual quality scores is presented in Table 1.

### 3.4. Results of Individual Studies

The findings of each study were summarized in descriptive matrices according to population, methodology, and main outcomes related to the open-door ICU model and humanized care. These matrices served as the foundation for the thematic synthesis (see Table 3, Table 4 and Table 5).

### 3.5. Results of Syntheses

The coding of qualitative data followed a systematic process of constant comparison, enabling the analysis of key relationships and concepts. This was conducted transparently to minimize bias and ensure the highest possible validity.

During open and axial coding, humanized care was defined through three fundamental dimensions:

Characteristics of the healthcare team include kindness, individualized care, dignity, ethical treatment, and respect for patient beliefs.

Communication between patients and professionals, including eye contact and clear explanations of procedures.

Comprehensive care provision of information, respect for privacy, flexible visitation, and family support.

Interactions between healthcare professionals and families were grounded in empathetic care across three core pillars: interaction, communication, and information—promoting mutual understanding and building relationships of trust and safety, which facilitate adaptation and coping for both patients and families.

Through selective coding, three main categories emerged: (1) Patient and Family Perceptions of ICU Care; (36% of the studies). (2) The need for flexible visitation and the use of technology to improve communication (30%). (3) Care centered on humanization (34%).

#### 3.5.1. Patient and Family Perceptions of ICU Care

ICU care is delivered by a multidisciplinary team committed to ensuring patient safety, efficiency, and quality of care for those in critical condition Hospitalization in the ICU is often a challenging experience for patients and families, frequently associated with emotional stress and anxiety [6,7,8,13,19,20].

Prolonged ICU stays can disrupt a person’s connection with their environment. Humanized care mitigates these effects by providing support that fosters adaptation and emotional well-being [21,22]. Patients and families may experience psychological and emotional consequences, including post-intensive care syndrome, when health needs are not met [23,24,25,26,27,28,29,30]. Conversely, continuous and compassionate engagement from healthcare professionals during hospitalization promotes trust, optimism, and a sense of emotional safety [31,32] (See Table 3).

#### 3.5.2. Flexible Visitation and Technological Mediation

The open-door ICU model allows patients to be surrounded by loved ones through flexible visitation policies, benefiting patients, families, and healthcare providers by fostering a more humanized and emotionally supportive environment. Family presence plays a critical role during ICU hospitalization, contributing to reduced length of stay, enhanced patient safety and confidence, and prevention of delirium and post-ICU psychological distress [3,4,5]. In contrast, the absence of a companion often leads to feelings of vulnerability, loneliness, and abandonment, particularly when restrictive visitation rules are in place [4,9,12,33,34].

Despite strong evidence supporting family involvement, some healthcare professionals continue to perceive family members as barriers to workflow or recovery, maintaining a preference for restricted visitation policies due to concerns about privacy, infection control, and perceived workload [33,34,35,36,37,38].

Information and communication technologies (ICT) have [36,37,38,39]. In the studies, no increase in adverse events, such as infections or delirium, was observed when comparing flexible visitation with traditional restrictive models.

Furthermore, respecting patient identity and personhood through personal items at the bedside, “Get to Know Me” boards, or spiritual care initiatives has been identified as a key component of humanized care that fosters connection, dignity, and trust between patients, families, and staff [19,36,40,41]. Nevertheless, the literature also highlights persistent organizational challenges, such as architectural limitations, noise levels, and emotional strain on staff, which influence the full adoption of open-door ICU policies and require supportive institutional strategies to overcome them [12,34,38,41,42] (See Table 4).

#### 3.5.3. Humanization-Centered Care

Humanization of care is a fundamental ethical principle. ICU admission places patients in a vulnerable position where identity and autonomy may be diminished. Care provided by the multidisciplinary team must therefore remain respectful and compassionate, regardless of the patient’s condition (e.g., intubated or sedated). Families also require accompaniment and education regarding the patient’s condition and care needs [1,2,10,11].

Findings underscore the importance of humanized care that respects dignity and personal values. This includes involving family members and using strategies to preserve patient identity, such as displaying personal photographs, allowing meaningful belongings (pillows, blankets, religious items), and fostering connections to the patient’s life story [14,15,43,44,45,46].

Effective humanized care demands trained staff capable of delivering timely, evidence-based, and compassionate interventions. Empathy and sensitivity are crucial for understanding patient suffering and improving well-being. Compassionate care addresses not only physical discomfort but also the psychological distress associated with loneliness, fear, loss of privacy, dependency, and uncertainty [47,48,49,50,51].

Healthcare professionals should be equipped with strong communication skills and human sensitivity, expressed through kind gestures, compassionate eye contact, gentle tone of voice, and active listening—essential elements to alleviate suffering and promote healing [52,53] (See Table 5).

### 3.6. Reporting Biases

No evidence of publication or selective reporting bias was identified among the included studies. All studies meeting the inclusion criteria were available in full text and provided sufficient methodological and contextual information to support data extraction and synthesis. The JBI appraisal revealed a generally low risk of methodological bias across the evidence base: 54% of studies were rated as high quality, 36% as moderate, and 10% as low quality. Minor sources of bias were related to incomplete methodological reporting in some narrative or integrative reviews and limited reflexivity statements in qualitative studies. In general, the findings indicate transparent reporting and robust methodological quality, supporting the credibility of the synthesized evidence.

### 3.7. Certainty of Evidence

No formal certainty assessment was conducted. Instead, confidence in the synthesized findings was appraised integratively by considering methodological quality (JBI scores), consistency of results across studies, adequacy of data, and direct relevance to the review question. Based on these criteria, the overall confidence in the evidence for each analytical theme was judged as follows:

Patient and family perceptions of ICU care: High confidence.

Supported by 18 studies (≈36%), predominantly qualitative and cross-sectional in design, with high methodological quality and consistent findings emphasizing communication, empathy, and emotional support as central elements of humanized care.

Flexible visitation and technological mediation: Moderate confidence.

Supported by 15 studies (≈30%), showing convergence between qualitative and quantitative findings on the benefits of family presence and digital communication tools. However, heterogeneity in study contexts and measurement methods reduced the overall certainty.

Care centered on humanization: High confidence.

Supported by 17 studies (≈34%), characterized by strong thematic coherence, high JBI quality scores, and consistent evidence underscoring the importance of ethical, relational, and compassionate dimensions in critical care.

Generally, the body of evidence demonstrates high confidence, reflecting methodological rigor, internal consistency, and clear alignment of the synthesized findings with the objectives of this review.

## 4. Discussion

Promoting a culture of open-door ICUs, consistent with humanization policies, contributes to the development of patient- and family-centered care. This approach requires valuing the psychosocial dimensions of health alongside the clinical condition, while upholding the ethical principles of intensive care. Many studies describe ICUs as hostile environments, with limited privacy and highly restrictive visitation policies. Therefore, strategies that ensure privacy and enable flexible visitation tailored to each family’s needs are crucial.

Regional differences in open-door ICU policies and humanized care strategies were evident. European countries such as Spain and the United Kingdom have adopted more standardized protocols for flexible visitation, whereas Latin American nations (e.g., Colombia, Chile, Ecuador) have implemented humanization initiatives in more variable ways, influenced by resource availability, cultural norms, and policy frameworks. Providing humanized ICU care requires advanced communication skills to convey complex technical information, manage expectations, and address the emotional responses of patients and families.

Multiple reviews indicate that flexible visitation reduces anxiety and stress in patients and families [30,31,32,34]. Nevertheless, many ICUs still maintain rigid visiting rules [23,50,53]. Swanson’s Theory of Caring underscores the social dimension of nursing, emphasizing that human connection is essential in all care settings [54]. However, for some healthcare professionals, family presence may generate anxiety, as it is perceived as an additional emotional and workload burden [22,23,42,53].

In many institutions, ICU facilities remain poorly adapted to the structural and operational requirements of an open-door model. A closed-door culture hinders meaningful interaction between patients and families [23]. While humanized care is widely recognized as a priority, its practical implementation remains challenging in highly complex units. Families, especially those experiencing the stress of critical illness, often desire prolonged contact and continuous updates on the patient’s status, which may conflict with the clinical need for controlled environments.

Effective communication between healthcare professionals and families has been shown to enhance satisfaction [22,33,36,37]. High-quality ICU practice requires advanced communication competencies to convey complex clinical information, align expectations with realistic outcomes, manage the psychosocial responses of patients and families, and deliver prognostic information in an ethically appropriate and professionally sensitive manner. During hospitalization, patients and relatives often develop strong bonds with nursing staff, seeking emotional support and opportunities to express concerns.

Mishel’s Uncertainty in Illness Theory [55] explains that uncertainty arises from the inability to interpret or predict illness-related events due to insufficient cues. This uncertainty can manifest as ambiguity about health status, complexity of treatment, lack of diagnostic information, or unpredictability of prognosis, each of which may provoke intense emotional responses. Nurses, through holistic and empathetic care, play a critical role in reducing this uncertainty by providing timely and clear information [8,20,21,23].

Communication challenges are particularly pronounced with intubated patients, who often experience frustration in expressing their needs. Healthcare providers may find it difficult to interpret these needs without dedicated time, interest, and the use of innovative non-verbal technologies [29,30,31].

This review also identified differences between adult and pediatric ICU family needs. Families of adult patients prioritize proximity and information as primary coping mechanisms [22,24,25]. In contrast, parents of pediatric patients, especially mothers, often wish to remain continuously present, even during invasive procedures, driven by the emotional imperative to protect their child [12]. Roy’s Adaptation Model [56] suggests that such responses reflect adaptive processes shaped by physiological and psychosocial factors. While adult patient families benefit most from effective communication, pediatric families require emotional support, transparent information, and active participation to foster healthy adaptation.

Cross-national differences further contribute to heterogeneity. In countries like Colombia, Mexico, and Chile, awareness of the importance of humanized care is growing, but implementation is often limited by cultural norms, insufficient training, and infrastructural constraints. In contrast, the United Kingdom, Switzerland, and Sweden have advanced open visitation policies, ICU diaries, and structured family support programs underpinned by public health policy and standardized protocols [4,23,38,40,57].

The studies included in this review varied widely in methodology. Qualitative research focused on lived experiences and perceptions [6,12,36], while quantitative studies measured outcomes such as satisfaction and anxiety reduction [30,31,32,34]. This diversity enriches the evidence base but limits direct comparability, underscoring the need for mixed-methods research to fully capture the complexity of ICU humanization.

Recent evidence reinforces the benefits of open-door policies. For instance, at a U.S. comprehensive cancer center, interventions such as extended visitation, communication boards, and improved waiting areas increased family inclusion (68.8% vs. 40.5%) and perceived support (48.8% vs. 29.1%) [20,32,33]. A cluster-crossover randomized trial comparing flexible (up to 12 h/day) versus restricted visitation (1.5 h/day) found no difference in infection or delirium rates but did report lower family anxiety and depression without increased staff burnout [42]. Similarly, a recent scoping review on pediatric ICU humanization strategies [44] found that while communication and end-of-life practices are improving, open visitation remains underdeveloped. These findings suggest that structural and communication interventions are not only safe and feasible but also effective across healthcare systems.

However, despite these benefits, potential challenges must also be considered. While open-door ICU models have been associated with reduced stress, anxiety, and delirium and, in some cases, with shorter ICU stays [12,24,27] several studies have reported concerns regarding infection control, increased staff workload, and potential distractions that may delay treatment or diagnosis, particularly in high acuity settings [41,42,58]. These findings highlight the importance of balancing humanization efforts with patient safety and clinical efficiency. Future research should explore strategies to mitigate these risks while sustaining the psychosocial and ethical benefits of humanized, family-centered intensive care.

### 4.1. Contextual and Methodological Considerations

The evidence synthesized in this review reveals considerable heterogeneity in regional contexts, patient populations, and health-system resource levels. Studies conducted in Europe and North America describe highly structured family engagement programs, supported by institutional protocols and national humanization frameworks. In contrast, investigations from Latin America and Asia highlight variable implementation of open-door ICU policies, often influenced by infrastructural limitations, cultural norms, and workforce constraints.

Despite these contextual differences, findings converge on the psychosocial benefits of flexible visitation and communication tools, including reduced anxiety and stress, improved understanding of the clinical condition, and greater satisfaction among patients and relatives, without any reported increase in adverse events such as infections or delirium.

Regarding population type, most studies focused on adult ICUs, while pediatric units emphasized the continuous emotional presence of parents as a critical component of family-centered and humanized care. These distinctions underscore the importance of tailoring open-door policies to the specific needs of each patient group and cultural setting.

Methodologically, the studies included span a wide range of designs from qualitative explorations of lived experiences to quantitative assessments of psychosocial outcomes reflecting both the complexity and adaptability of the open-door ICU model across diverse healthcare systems.

### 4.2. Implications and Recommendations for Practice

Based on the findings, institutions are encouraged to develop and implement ICU policies that facilitate comprehensive social and psychological support for patients and families. This may include structured family communication programs, flexible visitation protocols, and interdisciplinary training focused on humanized and family-centered care practices.

When care intentionally addresses emotional responses, uncertainty, and adaptation to critical illness, healthcare professionals are better equipped to meet both the physical and psychosocial needs of patients. A humanized model prioritizes dignity, autonomy, and well-being, aligning with the ethical principles of person-centered intensive care.

Environmental interventions such as reducing noise, optimizing lighting, preserving privacy through appropriate clothing, and creating a welcoming atmosphere further strengthen this approach. Allowing the use of mobile devices can also foster social connection and a sense of control, which are vital during prolonged ICU stays. Implementing these measures may enhance patient and family satisfaction, reduce suffering, and reinforce the ethical and humanistic foundation of ICU practice.

### 4.3. Strengths and Limitations

This review has limitations. First, publication bias may have favored studies reporting positive outcomes, potentially overrepresenting the benefits of open-door ICUs and humanized care. Second, restricting the search to English, Spanish, and Portuguese may have excluded relevant studies in other languages. Third, the seven-year search window (2018–2025) may have omitted earlier or more recent evidence. Fourth, most included studies were qualitative or descriptive, with only four randomized controlled trials and 54% rated as high methodological quality, which limits the generalizability of the findings. Fifth, the diversity of methodologies ranging from interviews to case studies and systematic reviews complicated direct comparisons and required interpretive synthesis, introducing potential subjectivity and data overlap. Finally, although the methodological quality and risk of bias were rigorously appraised using the Joanna Briggs Institute (JBI) Critical Appraisal Tools, no formal meta-analysis or quantitative certainty assessment was conducted. Despite these constraints, this review offers a comprehensive synthesis of current international evidence and highlights the growing global trend toward the humanization of intensive care.

## 5. Conclusions

This review identified key factors that influence the delivery of humanized care within the context of an open-door ICU model. Central to this approach is the active involvement of family members, even during critical illness. When supported by effective communication strategies and flexible visitation policies, this presence contributes to emotional support, trust-building, and improved adaptation to the ICU experience, while allowing families to participate in patient care with respect for privacy and dignity.

Preserving patient identity, for example, addressing patients by name even when intubated, reinforces recognition of the person behind the illness. The findings underscore the importance of a person-centered care model that addresses not only physical needs but also emotional, spiritual, and relational dimensions. Patients are emotional beings with individual preferences, and their ICU stay should increasingly resemble a humane and comforting environment characterized by active listening, empathy, and compassion.

Implementing open-door ICU policies in line with humanized care principles represents a necessary cultural shift. Moving toward this model has the potential to improve patient outcomes, strengthen family relationships, and enhance satisfaction with care, ultimately transforming the ICU into a more inclusive and healing space.

Despite methodological and contextual differences among the included studies, this review provides a comprehensive synthesis of current evidence on open-door ICU models and humanized care. It highlights the importance of balancing patient safety with emotional and ethical dimensions of care, promoting family involvement, and strengthening communication competencies among professionals. Together, these insights reflect a growing global trend toward the humanization of intensive care.

Future research should incorporate objective indicators such as ICU length of stay, infection rates, and mortality outcomes to more accurately assess the clinical benefits and potential risks of open-door ICU models, ensuring that humanization strategies are both compassionate and evidence-based.

## Figures and Tables

**Figure 1 nursrep-15-00406-f001:**
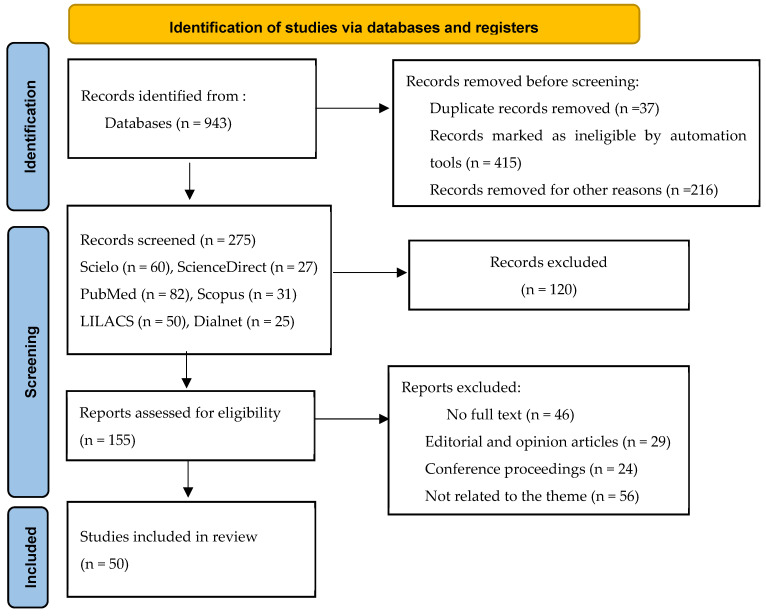
PRISMA 2020 Flow Diagram.

**Table 1 nursrep-15-00406-t001:** Methodological quality and risk of bias assessment by study design (JBI criteria).

Study Design	n	JBI Checklist Used	Score (%)	Quality Level	Risk of Bias
Review-based (integrative, narrative, systematic, scoping)	19	JBI Review-based	81%	Moderate–High	Moderate
Cross-sectional (descriptive/analytical)	15	JBI Cross-Sectional	87%	High	Low
Qualitative (phenomenological, descriptive, interpretive)	9	JBI Qualitative	88%	High	Low
Randomized Controlled Trials (RCTs)	4	JBI RCT	86%	High	Low
Cohort/Longitudinal	1	JBI Cohort	80%	Moderate	Moderate
Quasi-experimental	1	JBI Quasi-experimental	86%	High	Low
Mixed methods	1	JBI Mixed-Methods	100%	High	Low
Total	50	-	86% (average)	-	-

Note: Quality levels were classified as High (≥85%), Moderate (70–84%), and Low (<70%) according to JBI Critical Appraisal guidance.

**Table 2 nursrep-15-00406-t002:** Distribution of included studies by country, year, and methodological approach.

Country/Region	Number of Studies (%)	Year of Publication (2018–2025)	Articles (%)	Research Approach/Design	Number of Studies (%)
Colombia	9 (18%)	2018	11 (22%)	Review studies (integrative, narrative, systematic, and scoping)	19 (38%)
Spain	8 (16%)	2019	4 (8%)	Cross-sectional studies	15 (30%)
Chile	5 (10%)	2020	6 (12%)	Qualitative studies (phenomenological, descriptive, or interpretive)	9 (18%)
Ecuador, USA, Brazil	10 (20%)	2021	10 (20%)	Randomized controlled trials (RCTs)	4 (8%)
United Kingdom, Iran, Peru	6 (12%)	2022	11 (22%)	Cohort or longitudinal studies	1 (2%)
Mexico, Jordan, Norway, Sweden, Switzerland, Africa, Australia, New Zealand, Taiwan, Israel, Canada, China	11 (22%)	2023	5 (10%)	Quasi-experimental design	1 (2%)
Multiple continents	1 (2%)	2024	2 (4%)	Mixed Methods	2 (4%)
-	-	2025	1 (2%)		

**Table 3 nursrep-15-00406-t003:** Evidence synthesis: Patient and family perceptions of ICU care.

Author–Year	Country	Design/Approach	Population/Setting	Intervention/Exposure	Primary Outcome(s)	Key Finding	Ref.
Córdoba Rojas D.N., 2020	Colombia	Integrative/narrative review	Neonates and families in NICU	Open-door neonatal ICU model	Parental participation, bonding	Active parental involvement favors neonatal recovery and strengthens family–caregiver relationships.	[6]
Fuentes-Fernández E. et al., 2018	Mexico	Phenomenological study	Nurses in adult ICU	Open-door adult ICU experiences	Perceptions of family presence	Nurses emphasized need for clear, honest, and timely communication and emotional support for families.	[7]
Benavente Sánchez S., 2022	Spain	Literature review	ICU survivors	Post-ICU recovery, diaries	Psychological outcomes	Non-pharmacological interventions such as ICU diaries and rehab reduce post-ICU stress.	[8]
Maloh H.I.A.A. et al., 2022	Jordan	Cross-sectional descriptive	ICU nurses	Open visitation policy	Nurses’ perceptions	Despite perceived benefits, many nurses opposed open visitation; dialog needed to address cultural barriers.	[13]
Ahmad S.R. et al., 2023	USA	Qualitative study	ICU patients and families	“Get to Know Me” board	Dignity, personalization	Understanding patients’ preferences and lifestyle preserves dignity and human connection.	[19]
Azoulay É. et al., 2024	***** Multinational	Cross-sectional	ICUs in 11 countries	Family-centeredness of care	Family involvement, staff well-being	Supporting staff mental health enhances family-centered care; ethical climate influences humanization.	[20]
Canchero-Ramírez A. et al., 2019	Peru	Non-experimental cross-sectional	Families of ICU patients	Family satisfaction survey	Communication, satisfaction	Families seek frequent and transparent updates on patient progress.	[21]
Nicolalde Rodríguez D.M. et al., 2022	Ecuador	Systematic review	ICU patients	Humanized-care practices	Patient perception	Communication, respect, and empathy improve patient experience.	[22]
Naef R. et al., 2021	Swiss	Mixed-methods evaluation	Family members of ICU patients	Nurse-led family support intervention	Satisfaction, psychological well-being	Family support increases satisfaction and emotional adaptation, especially in end-of-life care.	[23]
Ceballos-Vásquez P. et al., 2021	Chile	Literature review	Families of ICU patients	Family participation in recovery	Family involvement	Families are essential to recovery despite visitation restrictions.	[24]
Gil J.B. et al., 2018	Spain	Exploratory study	Patients and relatives	ICU stay experience	Communication, environment	Improved communication and emotional support enhance satisfaction with ICU stay.	[25]
Schofield-Robinson O.J. et al., 2018	United Kingdom	Systematic review	ICU survivors	Follow-up programs post-ICU	Long-term well-being	Follow-up services improve quality of life and reduce post-ICU psychological distress.	[26]
Candal R.E. et al., 2024	USA	Cross-sectional	Surgical ICU staff	Visiting-hour preferences	Staff perceptions	Open visitation favored for family well-being without compromising care.	[27]
Duque-Ortiz C., Arias-Valencia M.M., 2020	Colombia	Integrative narrative review	ICU nurses and families	Nurse–family relationship	Humanized interaction	Strong nurse–family relationships enhance satisfaction and continuity of care.	[28]
Boada Quijano L.C., Guáqueta Parada S.R., 2019	Colombia	Quantitative study	Families of ICU patients	Information needs assessment	Communication quality	Continuous, clear communication reduces family anxiety.	[29]
Castillo J.M.V., Lagos Z.E.S., 2019	Chile	Quantitative cross-sectional	Family members of ICU patients	Family needs evaluation	Support, communication	Emotional support and effective communication improve family coping.	[30]
Padilla-Fortunatti C. et al., 2018	Chile	Descriptive/comparative quantitative	Families of critical patients	Family needs survey	Psychological burden	Meeting family needs decreases stress and uncertainty.	[31]
Abdul Halain A. et al., 2022	United Kingdom	Scoping review	Families of ICU patients	Psychological distress	Anxiety, depression, stress	Families experience high anxiety and emotional strain, impacting participation in care.	[32]

* Multinational study.

**Table 4 nursrep-15-00406-t004:** Evidence synthesis: Flexible visitation and technological mediation.

Author–Year	Country	Design/Approach	Population/Setting	Intervention/Exposure	Primary Outcome(s)	Key Finding	Ref.
Alonso-Rodríguez A. et al., 2021	Spain	Cross-sectional descriptive study	ICU nursing professionals	Open visitation policy	Professional perceptions, family interaction	Nurses recognized emotional benefits for families but expressed concerns about workload, infection risk, and privacy.	[3]
Bailey R.L. et al., 2022	Australia and New Zealand	Cross-sectional survey (WELCOME-ICU study)	ICU staff across 100 hospitals	Family access and visitation policies	Staff perceptions, perceived risk	Staff valued family access but reported stress and the need for clearer institutional guidelines to support open-door practices.	[4]
Guáqueta Parada S.R. et al., 2021	Colombia	Integrative review	Families of critical care patients	Nursing information interventions	Family information satisfaction	Effective information-sharing interventions improve family understanding and satisfaction during visitation.	[5]
Akbari R. et al., 2020	Iran	Randomized clinical trial	Adult ICU patients	Flexible visiting hours (vs restricted)	Physiological stability, anxiety	Flexible visiting significantly improved physiological stability and reduced anxiety in ICU patients.	[9]
Franchi R. et al., 2018	Ecuador	Cross-sectional descriptive study	Parents of pediatric ICU patients	Open-door pediatric ICU	Parental perceptions, satisfaction	Parents valued continuous presence during care and decision-making; emphasized emotional reassurance.	[12]
Quille-Manobanda D.N., Chipantiza-Barrera M.V., 2023	Ecuador	Qualitative phenomenological study	Family members of ICU patients	Open-door care experience	Communication, trust	Families perceived improved trust and connection through open visitation and active listening.	[33]
Hasandoost F. et al., 2023	Iran	Qualitative study	ICU staff and families	Humanistic care perception under open-door model	Family involvement, communication	Participants reported contradictory feelings—valuing closeness yet fearing loss of professional boundaries.	[34]
González-Martín S. et al., 2022	Spain	Randomized clinical trial	Families of preoperative cardiac surgery patients	Preoperative ICU visit program	Anxiety, depression, satisfaction	Short, structured ICU visits before surgery reduced family anxiety and increased satisfaction without adverse effects.	[35]
Jaramillo I.C., Zambrano G.I., Balda H.Z., 2021	Ecuador	Descriptive quantitative study	ICU health professionals	Open-door ICU policy	Professional attitudes	Staff perceived family presence as positive for care continuity but noted logistical and privacy challenges.	[36]
Sánchez-Alfaro L.A. et al., 2022	Colombia	Qualitative interpretive study	ICU healthcare professionals	Humanization in critical care practice	Meanings of care, family interaction	Professionals linked humanization to communication and flexible family participation in care.	[37]
Kvande M.E., Angel S., Nielsen A.H., 2022	Chile	Scoping review	International ICU settings	Humanizing intensive care (HumanIC)	Review of communication and visiting practices	Emphasized humanizing communication and visitation policies as central to person-centered ICU care.	[38]
Ardila-Suárez E.F., Arredondo-Holguín E.D.S., 2021	Colombia	Integrative review	Families in adult ICU	Nursing activities for family needs	Satisfaction, communication	Structured family visiting schedules and education improved satisfaction and reduced uncertainty.	[39]
Glimelius Petersson C. et al., 2018	Sweden	Comparative cohort study (2-month follow-up)	ICU patients and relatives	ICU diaries and family follow-up	Emotional recovery, memory integration	Diaries supported emotional recovery and communication between patients and their families post-ICU.	[40]
Palacio Jiménez M., 2020	Norway	Narrative review	ICU patients	Stress management in critical care	Anxiety, stress	Identified flexible visiting and emotional support as key to reducing patient stress.	[41]
Sanz-Osorio M.T. et al., 2023	Spain	Scoping review	Acute care units	Humanization of care and communication	Organizational practices	Highlighted the importance of clear communication and family inclusion for humanized environments.	[42]

Note: Each study appears only once, in the most relevant thematic category (Flexible visitation and technological mediation). Cross-references to related findings are provided in the main text.

**Table 5 nursrep-15-00406-t005:** Evidence synthesis: Care centered on humanization.

Author–Year	Country	Design/Approach	Population/Setting	Intervention/Exposure	Primary Outcome(s)	Key Finding	Ref.
Chang P.Y. et al., 2018	Taiwan	Cross-sectional correlational study	Primary family caregivers in ICU	Social support and stress-related symptoms	Stress, coping, support	Identified need for stress-coping interventions, as perceived social support did not reduce symptoms effectively.	[1]
Correa-Pérez L., Chavarro G.A., 2021	Colombia	Non-systematic literature review	Critical care patients	Comprehensive humanization framework	Care quality, ethical approach	Highlights the role of integrative bundles (e.g., ABCDEF) in improving outcomes and reducing adverse events.	[2]
Carpio Ahuana J., 2022	Peru	Cross-sectional relational study	ICU patients and relatives	Humanized nursing care vs. dependency level	Perception of care, dependency	Families perceived nursing care as moderately humanized; dependency level increased need for empathetic support.	[10]
Bosch L. et al., 2023	Spain	Pilot feasibility study	ICU patients, relatives, and staff	Implementation of ICU diaries	Communication, understanding	Diaries were feasible and improved empathy, communication, and emotional understanding.	[11]
Joven Z.M., Guáqueta Parada S.R., 2019	Colombia	Descriptive quantitative study	Critically ill patients	Nursing humanized care behaviors	Patient perception	Patients valued communication, professional presence, and empathy as indicators of humanized nursing.	[14]
Wang S. et al., 2020	China	Randomized controlled trial	Cardiac surgery ICU survivors	ICU diary intervention	Psychiatric symptoms, QoL, sleep	ICU diaries improved sleep quality and reduced anxiety and PTSD symptoms among survivors.	[15]
Perão O.F. et al., 2021	Brazil	Qualitative descriptive study	Families in palliative ICU care	Comfort and humanized support	Family perceptions of comfort	Comfort linked to empathy, proximity, and effective communication between team and family.	[43]
Salgado-Reguero M.E. et al., 2025	Spain	Scoping review	Pediatric ICUs	Humanization strategies	Implementation, feasibility	Pediatric ICUs apply humanization through communication, play, and family inclusion, but integration remains partial.	[44]
Garzón-Leguizamón L.F. et al., 2021	Colombia	Narrative review	ICU patients and professionals	Humanization as part of integral care	Empathy, professional training	Emphasizes training in empathy and closing the gap between humanization policy and clinical practice.	[45]
Sviri S. et al., 2019	Israel	Controlled pre–post quasi-experimental study	ICU family members	Structured communication tool	Family satisfaction, expectations	Structured communication improved satisfaction and clarified expectations regarding prognosis and care goals.	[46]
Petrinec A.B., Martin B.R., 2018	USA	Prospective longitudinal cohort	Family decision-makers of ICU patients	Post-ICU syndrome (PICS-F) assessment	Mental health, HRQoL	High prevalence of PICS-F associated with lower quality of life and avoidant coping among relatives.	[47]
Sili E.M. et al., 2023	Angola	Qualitative descriptive study	ICU nursing professionals	Humanized care discourse	Family inclusion, personalization	Emphasizes the need to extend humanized care to families through personalized, trust-based interactions.	[48]
Dos Santos E.L. et al., 2018	Brazil	Qualitative study	ICU nurses	Humanized assistance perception	Nurse perceptions	Nurses identified empathy, respect, and active listening as essential components of humanized care.	[49]
Espinoza-Caifil M. et al., 2021	Chile	Integrative review	ICU patients and nurses	Communication in critical care	Emotional care, communication	Emotional communication is often overlooked; improving it is key for humanized nursing.	[50]
Mussart K.M. et al., 2024	Brazil	Qualitative descriptive study	Families and ICU staff	ICU diaries implementation	Family connection, trauma prevention	Diaries strengthened emotional connection, prevented trauma, and supported family mental health.	[51]
Serrano García P., 2018	Spain	Narrative review	ICU nursing practice	Shift toward humanized nursing	Identity, patient perception	Discusses how humanized care transforms perceptions of illness and body identity in ICU settings.	[52]
Galvin I.M. et al., 2018	Canada	Systematic review	Healthcare professionals and relatives	Humanization of critical care	Psychological effects	Identifies emotional benefits of humanization for staff and relatives; bundles like ABCDEF improve safety and satisfaction.	[53]

## Data Availability

No new data were created or analyzed in this study. Data sharing is not applicable to this article.

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
