# Peer review of "Open-Door ICU Model and Humanized Care: A Systematic Review"

_nursrep, 2025, doi:10.3390/nursrep15110406_

Round 1
Reviewer 1 Report
Comments and Suggestions for Authors
< !--a=1-->< !--a=1-->

Author Response
We are very grateful for your thoughtful review and valuable observations. Your comments greatly contributed to improving the coherence, analytical depth, and presentation of our study. The following section outlines our detailed responses.

Reviewer 2 Report
Comments and Suggestions for Authors
1) Good concept, Well-written review about open ICU studies. This study highlights the humanitarian aspect of the open ICU model. If data is available, consider discussing other metrics like mortality or ICU stay, or infection rates in open ICU studies, or you can mention in the conclusion about including them in future studies to address the benefits and risks of changing the ICU model.
3.1 Study selection: Add 6 databases in parentheses next to 275 records in line 260. remove them in line 257.
Abstract: "Traditional intensive care units (ICUs) are often characterized by restricted visitation and a highly technological environment, which may contribute to patient de- personalization and emotional distress"- Management of patients in closed-door ICUs is often associated with limited family visits and a highly technological environment, which can lead to patient deconditioning through altered circadian rhythms and depersonalization, contributing to psychological distress in addition to physiological distress. "In recent years, there has been a growing global movement toward more humanized and family-centered care models, such as the open- door ICU, aiming to address these limitations." Rephrase- In recent years, there has been a shift in trend in the management of ICU patients with an emphasis on more social and psychological support, with the option of an open-door ICU. Objective: "To identify the key factors that facilitate or hinder the provision of humanized care in the context of the implementation of the open-door Intensive Care Unit (ICU) model, according to the perception of health professionals, patients, and their familiesRephrase- This study aims to evaluate the role of humanized care through social and psychological support in improving patients' outcomes through the concept of open-door ICU. Conclusions: Evidence highlights the importance of adopting humanized care practices in open door ICUs. Strategies like flexible visitation, emotional and spiritual support, respectful communication, and family involvement contribute to compassionate, patient-centered care. Findings underscore the need for institutional policies that support humanization and adaptation for patients and families. Rephrase: Based on the results of our systematic review, we emphasize the importance of adopting humanized care practices in open door ICUs. In particular strategies like flexible visitation, emotional and spiritual support, respectful communication, and family involvement contribute to compassionate, patient-centered care. We recommend institutional policies that need to be designed that support humanization for patients and families. Introduction-" The concept of an Open Intensive Care Unit (Open ICU), known in Spanish as UCI de puertas abiertas, reflects a growing international movement toward more humanized and family-centered critical care"- rephrase this is not scientific terms. The authors needs to write an article using scientific terms are described above instead of layman terms "These units promote attentive and compassionate care, in contrast to the traditionally rigid and highly regulated ICU environments that focus primarily on technological interventions and mortality risk"- can authors elaborate how attentive and compassionate care can be achieved based on open ICU by providing scientific evidence by citing the relevant articles. "Although modern ICUs have evolved to provide more personalized and continuous care, they are often perceived as unwelcoming due to factors such as intense lighting, alarm noise, and frequent staff conversations"- This could be a biased statement because the current ICU focuses on multiple factors to prevent postoperative delirium by minimizing the external factors. Restricted visitation and the inability of family members to accompany patients often lead to stress and a perceived loss of control-agree and this should be the article's main focus. "It seeks to achieve therapeutic goals without causing unnecessary"- sentence is incomplete "Implementing an open-door ICU model represents a significant shift toward this vision of humanized care by promoting transparency, emotional support, and family inclusion"- not scientific, rephrase. Methodology and results sections have been detailed and well described. The strongest part of this paper has been the analysis of bias. Discussion " Promoting a culture of open-door ICUs in alignment with humanization policies is essential to place patients and families at the center of care"-rephrase "Regional differences in open-door ICU policies and humanized care strategies were evident. European countries such as Spain and the United Kingdom have adopted more standardized protocols for flexible visitation, while Latin American nations ( have implemented humanization initiatives in more variable ways, influenced by resource availability, cultural norms, and policy frameworks"- rephrase ICU care requires advanced communication skills to deliver complex technical information, manage expectations, address emotional reactions, and "convey bad news" with clarity and compassion- These are not scientific words, the authors should use scientific terms- rephrase Implications and Recommendations for Practice- can authors make strong recommendations based on their study such as institutions developing ICU policies to facilitate the social and psychological support?
Recommendations-
The authors focused only on the emotional aspect. It would have been more informative if the authors actually reported based on the results of their systematic review that the impact of humanized touch in- 1. Reducing the duration of ICU stay. 2 Reduced incidence of ICU delirium The authors never emphasized the role of the Open ICU model in increasing the risk of infection, delay in the treatment. The authors missed the critical aspect of ICU care where delay in diagnosis and treatment with facilitating patient visitation may be associated with increased mortality without treatment.
1. The authors should include the factors that hinder the implications of open door ICU strategy and recommendations to overcome it . 2. The risks associated with open door ICU strategies. You can review the K.A.Milner et al 2020 multi center qualitative study which highlights the barriers of open door ICU.
Several limitations of the study:
Other limitation is the search criteria limited to only 7 years.
Only 54% of the studies are of high quality.
Only 4 out of 50 studies are Randomized controlled trials
Some of the studies included are systematic reviews
Open door ICU concept is not applicable in clinical practice needing further research and also can be harmful with risk of increased mortality.
Good luck!
Author Response

(The authors gave the same response as above.)
